# Platelet Activation Favours NOX2-Mediated Muscle Damage in Elite Athletes: The Role of Cocoa-Derived Polyphenols

**DOI:** 10.3390/nu14081558

**Published:** 2022-04-08

**Authors:** Alessandra D’Amico, Elena Cavarretta, Chiara Fossati, Paolo Borrione, Fabio Pigozzi, Giacomo Frati, Sebastiano Sciarretta, Vincenzo Costa, Fabrizio De Grandis, Antonia Nigro, Mariangela Peruzzi, Fabio Miraldi, Wael Saade, Antonella Calogero, Paolo Rosa, Gioacchino Galardo, Lorenzo Loffredo, Pasquale Pignatelli, Cristina Nocella, Roberto Carnevale

**Affiliations:** 1Department of Movement, Human and Health Sciences, University of Rome “Foro Italico”, 00135 Rome, Italy; a.damico@studenti.uniroma4.it (A.D.); chiara.fossati@uniroma4.it (C.F.); paolo.borrione@uniroma4.it (P.B.); fabio.pigozzi@uniroma4.it (F.P.); 2Department of Medical-Surgical Sciences and Biotechnologies, Sapienza University of Rome, 04100 Latina, Italy; elena.cavarretta@uniroma1.it (E.C.); giacomo.frati@uniroma1.it (G.F.); sebastiano.sciarretta@uniroma1.it (S.S.); antonella.calogero@uniroma1.it (A.C.); p.rosa@uniroma1.it (P.R.); 3Mediterranea, Cardiocentro, 80122 Napoli, Italy; mariangela.peruzzi@uniroma1.it (M.P.); pasquale.pignatelli@uniroma1.it (P.P.); 4IRCCS Neuromed, Località Camerelle, 86077 Pozzilli, Italy; 5AS Roma Football Club, Piazzale Dino Viola 1, 00128 Rome, Italy; vincenzo.costa@asroma.it; 6Villa Stuart Sport Clinic, FIFA Medical Center of Excellence, 00135 Rome, Italy; fabrizio_dg@libero.it (F.D.G.); antonianigro@tiscali.it (A.N.); 7Department of Clinical, Internal, Anesthesiological and Cardiovascular Sciences, Sapienza University of Rome, 00161 Rome, Italy; fabio.miraldi@uniroma1.it (F.M.); wael.saade@uniroma1.it (W.S.); lorenzo.loffredo@uniroma1.it (L.L.); 8Policlinico Umberto I, Viale Regina Elena, 328, 00161 Rome, Italy; gioacchino.galardo@uniroma1.it

**Keywords:** oxidative stress, exercise, platelet activation, polyphenols

## Abstract

Mechanisms of exercise-induced muscle injury with etiopathogenesis and its consequences have been described; however, the impact of different intensities of exercise on the mechanisms of muscular injury development is not well understood. The aim of this study was to exploit the relationship between platelet activation, oxidative stress and muscular injuries induced by physical exercise in elite football players compared to amateur athletes. Oxidant/antioxidant status, platelet activation and markers of muscle damage were evaluated in 23 elite football players and 23 amateur athletes. Compared to amateurs, elite football players showed lower antioxidant capacity and higher oxidative stress paralleled by increased platelet activation and muscle damage markers. Simple linear regression analysis showed that sNOX2-dp and H_2_O_2_, sCD40L and PDGF-bb were associated with a significant increase in muscle damage biomarkers. In vitro studies also showed that plasma obtained from elite athletes increased oxidative stress and muscle damage in human skeletal muscle myoblasts cell line compared to amateurs’ plasma, an effect blunted by the NOX2 inhibitor or by the cell treatment with cocoa-derived polyphenols. These results indicate that platelet activation increased muscular injuries induced by oxidative stress. Moreover, NOX2 inhibition and polyphenol extracts treatment positively modulates redox status and reduce exercise-induced muscular injury.

## 1. Introduction

Physical activity is considered a fundamental requirement for maintaining and promoting human health. Indeed, according to the World Health Organization (WHO), regular physical activity is useful in preventing and managing non-communicable diseases such as heart disease, stroke, diabetes and several cancers and also in counteracting hypertension, maintaining healthy body weight and last improving mental health and overall quality of life [1]. In this regard, athletic involvement may be a benefit to general, physical and mental health perception [2]. However, differences in the level of competition, frequency, and volume of exercise training, such as between amateurs and elite athletes, can impact differently the quality of life with different consequences for athletes in terms of incidence of illness and injury. Indeed some studies indicate that long-term regular and moderate practice of aerobic physical activity exerts a positive impact by enhancing the immune function response, reinforcing the anti-oxidative capacity and increasing the efficiency of energy generation, finally reducing the incidence of inflammatory diseases [3,4]. Conversely, intense physical activity may induce free radical generation, inflammatory responses, and hormonal and biochemical changes that can favours muscle damage and dysfunction [5,6], key components of musculoskeletal injuries [7].

Specifically, musculoskeletal injuries represent a significant percentage of all traumas in sports medicine, with an incidence ranging from 10% to 55% of all experienced injuries. The extent of muscle damage, defined as a prolonged dysfunction that needs several days for a complete recovery [7], depends on the mode, intensity, duration of exercise and prior exercise training. In the case of severe muscle damage, the injury can result in a reduction of physical activities and may interfere with the training and competitive schedules of athletes [8].

Among factors playing a key role in skeletal muscle pathophysiology, oxidative stress is associated with important molecular, structural, and functional implications [9]. Excessive oxidative stress is stimulated by exercise. The increase in reactive oxygen species (ROS) production can be dependent on the type, duration, and load of the exercise [10]. At low to moderate levels, ROS act as signal molecules promoting the production of enzymes relevant to the adaptation of muscle cells to intense exercise, eliciting positive effects on muscle physiological responses. Conversely, during intense and prolonged exercise, skeletal muscle becomes the major source of ROS leading to excessive oxidative stress that can result in impaired physical performance and maladaptive skeletal muscle recovery [11]. The upregulation of NOX2, the catalytic subunit of NADPH oxidase, plays a key role in the formation of ROS [12]. NOX2 produces superoxide anion, which is rapidly dismutes into H_2_O_2_, further amplifying intracellular oxidative stress [12].

Platelet function is directly and indirectly affected by physical activity or inactivity. Indeed, a large body of evidence indicates that both acute exercise and continuative physical activity affect platelet function [13]. In particular, exercise intensity represents a critical determinant of platelet activation as indicated by increased platelet adhesion [14], increased intraplatelet Ca^2+^ levels [15], and platelet aggregation in response to several agonists [14,16,17].

Several mechanisms are supposed to contribute to exercise-induced platelet activation. For example, exercise is characteristically accompanied by sympathetic nervous system (SNS) activation with a rise of plasma catecholamines [18,19] that result in increased platelet aggregation through platelet α_2_-adrenergic receptors [20]. Moreover, when activated, platelets release a variety of proteins largely stored in the characteristics granules including alpha granules, dense granules and lysosomes [21]. The repertoire of soluble mediators released mediates, in addition to haemostasis, functional responses contributing to both inflammation and thrombosis, eventually leading to thrombo-inflammatory activities in vascular diseases [22].

However, the role of platelet activation, and specifically the role of molecules released upon activation, on muscle damage are still debated.

Accordingly, purposes of this study were to (1) compare platelet activation, oxidative biomarkers and muscle damage induced by physical activity in amateurs and elite athletes; (2) explore if platelet activation and oxidative stress could lead to muscle damage; (3) verify if natural molecule with scavenging activity, i.e., polyphenols, could help in reducing muscle damage in vitro.

## 2. Materials and Methods

### 2.1. Study Population

The study was performed on 23 elite male Football players (30.1 ± 4.8 years). Elite athletes were all practicing football that can be classified as a highly intensive intermittent sport [23]. Players had at least 12 years of previous football training and were all primary division (Italian “Serie A” team) members of the first-league A.S. Roma Football team, they trained at least 14 h/week, >6.0 METs and had at least 8 years of competitive experience. At the time of enrolment, the athletes were enrolled after 45 days off-season holidays in which they didn’t carry out specific activities such as football, but only recreational sports fitness. During the training and competitive seasons, elite athletes were engaged in a 120-min training (including a 15-min warm-up, 30-min technical-tactical skills, 30-min aerobic training reaching 75% of the maximal heart rate, 30-min strength training, and 15-min cooldown) 6 times per week and a 90-min match per week.

Exclusion criteria were the occurrence of musculoskeletal injuries in the last month before enrolment and assumption of drugs or supplements that could influence the experimental protocol.

Amateur athletes were age- and sex-matched with 23 physically active male subjects (30.2 ± 4.7 years), engaging in at least 3-day·week-1 of moderate-to-intense physical activity, ranging from 3.0 to 6.0 METs and/or >6.0 METs who practice mixed sports disciplines except football, specifically 14 (61%) athletes played tennis/padel tennis, 8 (35%) athletes practiced high-intensity interval training and 1 (5%) practiced surfing. Only athletes practicing mixed sports were included, because mixed sports fall in the same category as football based on the Dal Monte Lubich sports classification [24], which is based on the energetic sources, characterized by an alternated aerobic-anaerobic impact and in the cardiovascular classification of the Olympic sport disciplines [25] football and the other disciplines included falls in the category of the mixed sports” which demonstrate a balanced cardiovascular remodelling, according to the relative isometric and isotonic components of exercise and resulting cardiovascular adaptation (Table 1).

#### 2.1.1. Blood Sampling and Preparations

All blood samples were collected in the morning (8–9 a.m.), from the antecubital vein in seated position in fasting athletes. Plasma and serum samples were collected in BD Vacutainer (Franklin Lakes, NJ, USA) with or without anticoagulant [trisodium citrate, 3.8%, 1/10 (*v/v*)], respectively. The blood was centrifugation at 300× *g* for 10 min at room temperature (RT). The supernatants were divided into aliquots and stored at −80 °C for analyses. Blood samples were collected prior to the beginning of the training season in elite athletes, at the same time of the pre-participation screening, to better normalized the reactive oxygen species production [26]. The training in the days before blood sampling was free for elite and amateur athletes, but all elite and amateur athletes abstained from training for 24 h before blood sampling, as requested by our study protocol. The training session was not standardized for the elite nor for the amateur athletes, as the elite athletes were enrolled immediately prior to the beginning of the training and competitive season, after 45 days off-season holidays in which they didn’t carry out specific activities such as football, but only recreational sports fitness.

#### 2.1.2. Evaluation of Platelet Activation Biomarkers

Plasma levels of soluble(s)CD40L were measured with the use of a commercial immunoassay (Boster Biological Technology) and expressed as pg/mL. Intra-assay and inter-assay coefficients of variation were 5% and 7%, respectively.

Plasma levels of PDGF-bb were measured with the use of a commercial immunoassay (Cusabio, Houston, TX, USA) and expressed as pg/mL. Intra-assay and inter-assay coefficients of variation were <8% and <10%, respectively.

#### 2.1.3. Evaluation of Catecholamine Levels

Human Dopamine Decarboxylase (DDC) was analysed by ELISA Kit (Elabscience, Houston, TX, USA). Values were expressed as pg/mL. Intra-assay and inter-assay coefficients of variation were <10%.

Epinephrine/Adrenaline levels were analysed by ELISA Kit (Fine Test, Wuhan, Hubei, China). Values were expressed as pg/mL. Intra-assay and inter-assay coefficients of variation were <8% and <10%, respectively.

#### 2.1.4. Evaluation of Oxidative Stress

Serum hydrogen peroxide (H_2_O_2_) breakdown activity (HBA) was measured with HBA assay kit (Aurogene, code HPSA-50). The % of HBA was calculated according to the following formula: % of HBA= Ac − As/Ac * 100 where Ac is the absorbance of H_2_O_2_ (1.4 mg/mL), As is the absorbance in the presence of the serum sample.

NOX2 activity was measured in serum and platelets as sNOX2-dp with a previously reported by ELISA method [27]. Values were expressed as pg/mL; intra- and inter-assay coefficients of variation (CV) were 8.95% and 9.01%, respectively.

The Hydrogen Peroxide (H_2_O_2_) was measured by using a colorimetric assay as described previously. A standard curve of H_2_O_2_ (0–200 μM) was performed for each assay. Briefly, 50 μL of cell supernatant was mixed with 3,3′,5,5′ tetramethylbenzidine (50 μL, Sigma-Aldrich, St. Louis, MI, USA) in 0.42 mol/L citrate buffer, pH 3.8, and 10 μL of horseradish peroxidase (52.5 U/mL, Sigma-Aldrich, St. Louis, MI, USA). The samples were incubated at room temperature for 20 min, and the reaction was stopped by the addition of 10 μL 18 N sulphuric acid. The reaction product was measured spectrophotometrically at 450 nm and expressed as μM.

#### 2.1.5. Evaluation of Muscle Damage

Muscle damage was evaluated by serum creatine kinase (CK), lactate dehydrogenase (LDH), and myoglobin levels analysed using a commercial ELISA kit (Antibodies-online GmbH, Aachen, Germany; EIAab Wuhan Hubei, China, China; DRG Instruments GmbH, Marburg, Germany).

### 2.2. In Vitro Studies

#### 2.2.1. Platelet Preparation and Aggregation

Blood samples added with sodium citrated (3.8%, 1/10 (*v/v*) were taken from healthy subjects (HS, n = 3) in fasting conditions. Blood was centrifuged for 15 min at 180× *g* at room temperature (RT) and PRP (2 × 10^5^ platelets/μL) was prepared as previously described.

Platelet aggregation was induced in PRP samples by collagen (2 µg/mL or 0.25 µg/mL) and was measured for 8 min. Before activation, samples were pre-incubated (20 min at 37 °C) with different concentrations of catecholamine.

Platelet aggregation experiments were performed by platelet aggregation profiler PAP-8E (Bio/Data corporation, Horsham, PA, USA), using Born method. Finally, samples were centrifuged for 3 min at 3000 rpm and supernatants and pellets were stored at −80 °C.

#### 2.2.2. Cell Culture and Reagents

The Human skeletal muscle myoblasts cell line (HSMM, Lonza, Basel, Switzerland) was cultured in SkBM^TM^-2 Basal Medium and SkGM^TM^-2 SingleQuots^TM^ supplements, 2 mM glutamine, and 1% antibiotics (all Gibco) for expansion and maintenance of the undifferentiating state. When cultures reached 80% confluence, myogenic differentiation was induced by replacing the expansion media with DMEM/0.2% FBS. Afterward, cells were pre-treated for 2 h either with vehicle (Phosphate Buffered Saline, PBS) or NOX2ds-tat (10 μM in PBS; Anaspec, Fremont, CA, USA), the most specific inhibitor of interactions between NOX2 and p47phox [28]. Cells were then stimulated with 10% plasma obtained from 3 different amateurs and 3 different elites’ athletes for 24 h.

Experiments were also conducted by treating cells either with vehicle (ddH_2_O) or cocoa-derived polyphenols (1 μM in ddH_2_O) for 1 h before stimulation with 10% plasma obtained from 3 amateurs and 3 elites’ athletes for 24 h. The cocoa-derived polyphenols concentration was chosen as the lowest non-toxic concentration to obtain the inhibition. The experiments were conducted on three different batches of HSMM.

Conditioned media were harvested and tested for the quantification of soluble NOX2 and H_2_O_2_ as described, and pellets were analysed in western blot to verify muscle damage with α-actin expression.

#### 2.2.3. Protein Detection, Electrophoresis, and Western Blot Analysis

Cell pellets were suspended in RIPA buffer (5 mM EDTA, 0.15 mol NaCl, 0.1 mol Tris pH 8.0, 1% Triton) with protease and phosphatase inhibitors cocktail (10 μg/mL; Thermo Fisher Scientific, Waltham, Massachusetts, USA) and sonicated three times (for 10 s and 70% amplitude). The samples were incubated on ice for 30 min, and then centrifuged at 10,000× *g* for 20 min to remove pellet residues. Equal amounts of protein (30 μg/lane) estimated by Bradford protein assay were solubilized in a 2X Laemmli sample buffer containing 20% of 2-mercaptoethanol. Proteins were separated by SDS-PAGE on 10–12% polyacrylamide gel and then electro-transferred to nitrocellulose membranes (Trans-Blot Turbo Mini Nitrocellulose, Bio-Rad, Hercules, CA, USA). Membranes were, then, incubated overnight at 4 °C with rabbit polyclonal anti-α-actin (Abcam, Cambridge, UK) and with mouse monoclonal anti-vinculin (Santa-Cruz Biotechnologies, Dallas, TX, USA). Thus, membranes were incubated for 1 h with HRP-conjugated secondary antibody (1:3000; Bio-Rad, CA, USA). The immune complexes were detected by enhanced chemiluminescence substrate (ECL Substrates, Bio-Rad, CA, USA). Densitometric analysis of the bands was performed using Image J software. The results were expressed as arbitrary unit (A.U.) and represented as mean of three independent experiments.

#### 2.2.4. Extraction of Phenolic Fraction from Chocolate and Total Polyphenol Content Evaluation

One gm of chocolate was weighed, and fat was removed by using 1 mL of n-hexane. Polyphenols were extracted from the defatted pellet using a total volume of 3 mL (1 × 3 mL) with 80% (*v/v*) of acetone:water at 80 °C. This aqueous acetone solution, which contained most of all polyphenols, was used for polyphenols analysis and for in vitro study.

#### 2.2.5. Extraction and Quantification of Catechin and Epicatechin from Chocolate

Catechin and epicatechin were extracted and quantified according to Gottumukkala et al. [29]. Briefly, about 10 g of chocolate sample was extracted with methanol on a hot water and all methanolic fractions were combined, filtered, and evaporated. Catechin and epicatechin stock solutions were prepared in methanol in the concentration range of 100–600 μg/mL. The HPLC analysis was performed using an HPLC system (Agilent 1200 Infinity series HPLC system, Santa Clara, CA, USA). The column temperature was maintained at 30 °C. HPLC mobile phase was solution A: 0.1 mL of orthophosphoric acid dissolved in 900 mL of HPLC grade water and Solution B: acetonitrile. Mobile phase was run using gradient elution: at the time 0.01 min 11% B; at the time 30 min 25% B; at the time 35 to 39 min 100% B; and at the time 40 to 50 min 11%B. The mobile phase flow rate was 1.0 mL/minute, and the injection volume was 10 μL. The eluents were detected and analysed at 280 nm.

### 2.3. Statistical Methods

#### 2.3.1. Categorical Variables Are Reported as Counts (Percentage) and Continuous Variables as Mean ± SD for Those without Normal Distribution and as Median (Interquartile Range [IQR]) for Continuous Variables without Normal Distribution

Comparison of categorical variables was tested by χ^2^ test and comparisons between groups for continuous variables were carried out by Student’s *t* test; in case of non-normal distribution the comparisons of the continuous variables were performed through non-parametric tests (Mann Whitney U test).

The Shapiro-Wilk test was used to determine the normality of the distribution. Simple linear regression analysis was performed by Spearman’s rank correlation test. *p* < 0.05 was considered statistically significant. All analyses were carried out with SPSS version 25 (SPSS Inc., Chicago, IL, USA).

#### 2.3.2. Effect Size and Sample Size Determination

The effect size of s-NOX2dp was defined as: mean of elite athletes − mean of amatorial athletes/pooled standard deviation. The effect size (Cohen’s d) was 1.9.

We computed the minimum sample size with respect to a two-tailed, one-sample Student’s t-test with Welch correction, considering, on the basis of data coming from a previous pilot study (data not shown): (i) a difference for s-NOX2dp levels to be detected between amateur and elites athletes δ: 6 pg/mL; (ii) s.d.: 6.2; (iii) type I error probability α: 0.05 and power 1 − β: 0.90; this resulted in n = 23 patients/group.

## 3. Results

### 3.1. In Vivo Study

#### 3.1.1. Exercise Induces Platelet Activation and Granule Release

In order to verify the degree of platelet activation according to physical activity, we evaluated platelet activation by the release of soluble mediators of preformed intracellular vesicles such as sCD40L, and PDGF-bb stored in *α*-granules.

Results showed that sCD40L levels are higher in the elite athletes’ group compared to amateurs (313.4 ± 73.96 pg/mL vs. 212.5 ± 48.8 pg/mL, *** *p* < 0.001) (Figure 1A). Differences were also found for PDGF-bb concentration with significant increased levels in the elite athletes’ group compared to amateur group (8.58 ± 3.1 ng/mL vs. 5.7 ± 1.35 ng/mL, ** *p* < 0.01) (Figure 1B).

During intense exercise, the sympathetic nervous system and the adrenal medulla produce and consequently release catecholamines directly proportional to the intensity of physical effort. Accordingly, the blood levels of the enzyme dopa decarboxylase, the key enzyme for the transformation of L-dopa into dopamine, were therefore analysed. Results showed that the highest levels were found in the elite athletes group compared to amateurs (26.17 ± 14.6 pg/mL elite vs. 10.0 ± 4.8 pg/mL amateurs, *** *p* < 0.001) (Figure 1C). Conversely, no difference was found in epinephrine concentration (data not shown). Finally, we found a positive correlation between levels of sCD40L and dopamine (Rs = 0.176; *p* = 0.0003) (Figure 1D).

#### 3.1.2. Intensive Exercise Induces Oxidative Stress

To quantify the levels of oxidative stress, we analysed H_2_O_2_ production and NOX2 activation. The highest levels of H_2_O_2_ were found in the elite athletes’ group compared to amateurs (23.07 ± 11.17 µM vs. 15.14 ± 5.013 µM, ** *p* < 0.01). Consistently with increased H_2_O_2_ production, we found the highest levels of sNOX2dp in the elite athletes’ group compared to the amateurs (8.141 ± 4.21 pg/mL vs. 2.02 ± 1.69 pg/mL, ** *p* < 0.01) (Figure 2A,B). The antioxidant status, evaluated by the measure of the percentage of H_2_O_2_ that is neutralized by antioxidant enzymes, was significantly reduced in the elite athletes’ group compared to the amateurs’ group (32.9 ± 6.5% vs. 48.0 ± 12.1%, *** *p* < 0.001) (Figure 2C).

#### 3.1.3. Intensive Exercise Induces Elevation of Specific Muscle Enzymes

The analysis of creatine kinase (CK), lactate dehydrogenase (LDH) and myoglobin showed higher muscle injury in the elite athlete group compared to amateurs (669.9 ± 160.3 mU/mL vs. 491.9 ± 62.02 mU/mL, *** *p* < 0.001 for CK, 159.3 ± 38.22 mU/mL vs. 127.4 ± 52.85 mU/mL, * *p* < 0.05 for LDH, and 156.9 ± 97.55 ng/mL vs. 104.5 ± 8.318 ng/mL, ** *p* < 0.01, for myoglobin) (Figure 3A–C).

As reported in Table 2, we found a positive correlation between biomarkers of muscle injury and oxidative stress and platelet activation.

### 3.2. In Vitro Study

#### 3.2.1. Dopamine Induces NOX2-Mediated Oxidative Stress and Platelet Aggregation

To evaluate if dopamine is able to increase oxidative stress and platelet activation, we treated platelets with dopamine in the range of concentration found in the serum of elite athletes. The results showed that dopamine (15–30 pg/mL) incubated with platelets in the presence of threshold (2 µg/mL) or sub-threshold (0.25 µg/mL) concentration of collagen significantly increased NOX2 activation, H_2_O_2_ production and platelet aggregation (Figure 4A–D).

#### 3.2.2. Plasma from Elite Athletes Increases Oxidative Stress and Muscle Injury: The Role of NOX2

To test the possible role of molecules released by activated platelets in muscle injury, human skeletal muscle myoblasts cell line was incubated with plasmas derived from amateurs and elite athletes.

The results showed that, when cells were treated with plasma from elite athletes, a significant increase in H_2_O_2_ production and NOX2 activation was observed compared to the treatment with amateurs’ plasma (Figure 5A,B). Moreover, plasma from elite athletes significantly increased CK, LDH and myoglobin (Figure 6A–C). The effect on muscle injury was also confirmed by the analysis of the expression of α-actin (Figure 6D).

Finally, to investigate the underlying mechanism, cells added with plasma were pre-treated with NOX2ds-tat. The results showed that the inhibitor significantly reduced H_2_O_2_ production, NOX2 activation (Figure 5A,B) and muscle injury elicited by plasma (Figure 6A–D).

#### 3.2.3. Cocoa-Derived Polyphenols Reduces Plasma-Induced Muscle Damage

Polyphenols are a large group of natural compounds found in food and beverages that exert antioxidant properties with a beneficial effect to reduce the rate of muscular injury in endurance athletes [30,31]. For this purpose, we performed in vitro study with cocoa-derived polyphenols such as catechin and epicatechin (Table 3). The results showed that cells treated with plasma, but pre-incubated with polyphenols, displayed a significant reduction of oxidative stress (Figure 7A,B) and muscle injury (Figure 7C,D) elicited by plasma.

## 4. Discussion

This study showed that: (a) platelet activation, oxidative stress and markers of muscle damage are significantly increased in elite football players compared to amateurs; (b) elite football players have increased dopamine concentration compared to amateurs; (c) dopamine plays a key role in favouring platelet activation and in promoting oxidative stress; (d) molecules released in plasma from elite athletes increase in vitro muscle injury and NOX2-mediated oxidative stress and (e) cocoa-derived polyphenols are able to exert a beneficial effect by reducing oxidative stress and muscle injury.

Physical exercise represents a major challenge to whole-body homeostasis promoting a wide range of biological perturbations in cells, tissues, and organs that are caused by the increased metabolic activity of contracting skeletal muscles [32]. Multiple integrated responses interplay to minimize the homeostatic perturbation generated by the increase in muscle energy and oxygen demand. This metabolic challenge produces an imbalance between antioxidant defences and the generation of reactive oxygen species (ROS) that are important modulators of muscle contraction, antioxidant molecules upregulation, and oxidative damage repair [33].

However, the relationship between exercise and oxidative stress is extremely complex and mainly depends on mode, frequency, intensity, and duration of exercise [33]. Indeed, low and physiological levels of ROS are required for normal force production in skeletal muscle kinetics, but high levels of ROS promote contractile dysfunction resulting in muscle injury. Several experimental and epidemiological studies have reported this double effect according to the levels of oxidative stress. Some studies have underlined the role of physical activity mainly associated with improved nitric oxide function by eNOS activation [34,35]. Other studies report that chronic regular training increases the production of ROS, particularly suggesting that high-intensity, regular training can cause redox imbalance overwhelming the antioxidant defence ability, leading to several types of injuries [36,37,38,39].

Accordingly, in this study, we found a different degree of oxidative stress according to physical activity intensity. Compared to amateurs, elite football players displayed increased oxidative stress as indicated by increased NOX2 activation and H_2_O_2_ production coincidentally with the reduced antioxidant property as indicated by decreased HBA.

During physical exercise, several mechanisms increased the generation of ROS such as mitochondrial superoxide/H_2_O_2_ production in skeletal muscle [11], and increased activation of enzymatic cellular source of ROS [26], including platelets [40]. In particular, studies of platelet function during physical activity are of special interest as there is controversial evidence on the effect of different types and workloads of physical exercise on platelet function [41]. Indeed it has been reported that acute and moderate-intensity exercise may reduce platelet activation [42], however, other studies observed acute platelet activation, aggregation and adhesion during physical exercise [43,44,45].

In this study, we showed that platelets derived from elite football players are more activated as indicated by increased sCD40L and PDFG-bb release, underlying a different contribution of intensive physical exercise in promoting platelet function.

Catecholamines are released during exercise and represent another mechanism involved in free radical production. Levels of circulating catecholamines may vary in response to different types of stimuli. Intensive physical exercise, in fact, promotes the release of greater quantities of adrenaline and dopamine. However, this phenomenon is strongly influenced by numerous factors including the type and duration of training. Our results showed higher blood dopamine concentrations in the elite athletes’ group than in the amateur group with a significant correlation with platelet activation, confirming a close relationship between dopamine and platelet function. Thus, we performed an in vitro study to test whether dopamine, at the concentrations found in the circulation of elite athletes, might be able to promote platelet aggregation and oxidative stress. Indeed, human platelets express dopamine D3 and D5 receptors [46]; moreover, catecholamines are concentrated in human platelets and stored in dense granules [18]. Although platelets lack enzymes for catecholamine synthesis, the platelet level of catecholamines is higher than in plasma. This catecholamine pool will be released during platelet secretion [19], acting in an autocrine and paracrine manner to amplify platelet activation. The results showed that dopamine levels, at the concentration found in circulation, can promote platelet aggregation and oxidative stress.

These alterations in oxidative status parameters and dopamine-induced platelet function can alter muscle cells leading to decreased muscle function, and performance. As expected, we confirmed an increase of muscle damage biomarkers such as CK, LDH, and myoglobin in elite athletes compared to amateurs with a significant correlation with oxidative stress and platelet activation biomarkers.

To further explore the contribution of oxidative stress and platelet activation in muscle injury, we performed in vitro study on cell cultures of human myoblasts, treated with plasma from elite athletes or amateurs. We speculated that plasma contained molecules and several effectors released from activated platelets coincidentally with increased oxidative stress effectors and that these molecules can alter muscle function contributing to the damage. Our results highlighted that plasma from elite athletes is able to increase oxidative stress markers (H_2_O_2_ and sNOX2-dp) and muscle injury markers (CK, LDH and myoglobin). Moreover, α-actin, a reliable marker of skeletal muscle damage [47], is significantly increased after myoblast cells treatment with elite athletes’ plasma. This effect was blunted when cells were pre-treated with NOX2ds-tat, that by inhibiting NOX2 activation, restores oxidative stress and muscle injury.

Controlling or minimizing muscle injury can promote faster recovery, maximize training and performance, and prevent injuries. Thus, supplementation with dietary compounds has been increasingly frequent to improve sports performance [48,49]. Scientific evidence has shown positive modulation of the redox status and a reduction of exercise-induced muscular injury biomarkers by polyphenol-rich nutrient supplementation [31]. However, the mechanism is not clarified. Thus, we incubated human myoblasts cells with cocoa-derived polyphenols (catechin and epicatechin) before the stimulation with the plasma of elite athletes and amateurs. We found a significant down-regulation of NOX2 activation, H_2_O_2_ production and reduction of CK and α-actin after cell treatment.

This study has some limitations. First, we have not performed a clinical evaluation of muscle damage. Therefore, the increase in muscle biomarkers only indicates a muscle injury and we only speculate that increased oxidative stress, and in particular NOX2 activation, could be predictive of any muscle damage. Second, the sample size is too small to evaluate the differences between different sport by amateur athletics. Third, previous studies [50,51,52] indicates that platelet biology and oxidative stress are influenced by gender. Specifically, in physiological conditions, platelet activation is enhanced in woman [50] that appear to be less susceptible to oxidative stress [51]. As these physiological differences could add a confounding factor, we only included men in this study. Fourth, we did not evaluate the role of platelet and NOX2 activation in muscle injuries in elite athletes who may have different training programs.

Finally, the effect of cocoa-derived polyphenols is only tested by in vitro studies. An interventional study with polyphenols enriched-nutrients supplementation would be necessary to support a polyphenols-based strategy to reduce muscle injury in athletes.

In conclusion, platelet activation plays a key role in intensive exercise-induced muscle injury and NOX2-mediated oxidative stress is a fundamental mechanism in the impairment of muscle cell integrity (Figure 8). Therefore, nutritional strategies using foods with antioxidant effects could reduce NOX2 activation and then could represent a good strategy to reduce muscle injury improving athletic performance (Figure 8).

## Figures and Tables

**Figure 1 nutrients-14-01558-f001:**
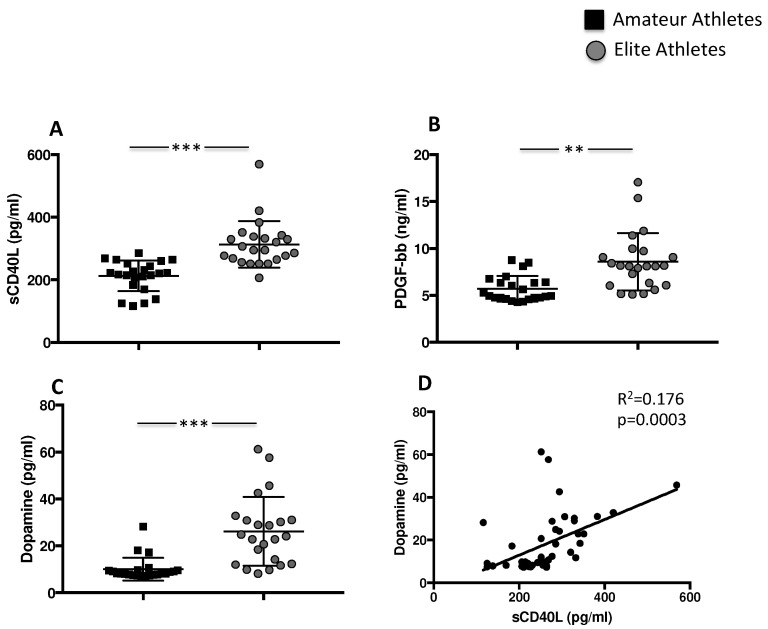
Platelet activation in amateurs and in elite football players. Serum levels of (**A**) sCD40L, (**B**) PDGF-bb and (**C**) dopamine in amateurs (n = 23) and in elite football players (n = 23) (Data are represented as mean ± standard deviation) (** *p* < 0.01; *** *p* < 0.001). Linear correlation between dopamine and sCD40L (**D**) in 23 amateurs and 23 elite football players.

**Figure 2 nutrients-14-01558-f002:**
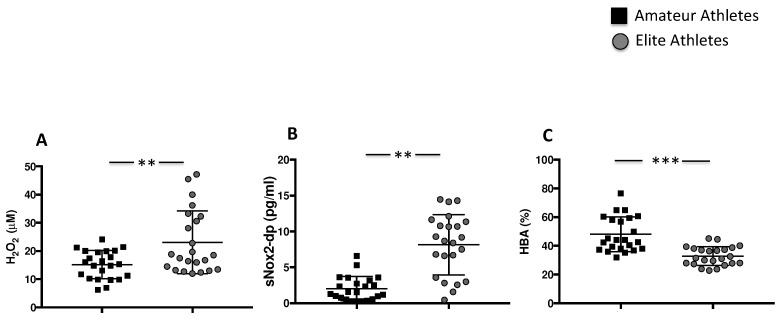
Oxidative stress in amateurs and elite football players. Serum levels of H_2_O_2_ (**A**), sNOX2-dp (**B**) and % of HBA (**C**) in amateurs (n = 23) and in elite athletes (n = 23). (Data are represented as mean ± standard deviation) (*** *p* < 0.001; ** *p* < 0.01).

**Figure 3 nutrients-14-01558-f003:**
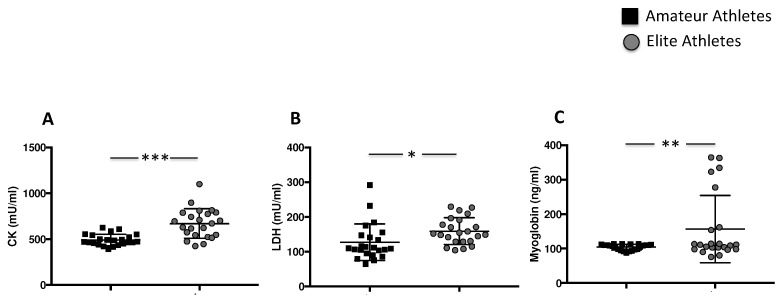
Muscle injury enzymes in amateurs and elite football players. Serum levels CK (**A**), LDH (**B**) and Myoglobin (**C**) in amateurs (n = 23) and in elite athletes (n = 23). (Data are represented as mean ± standard deviation) (*** *p* < 0.001; ** *p* < 0.01; * *p* < 0.05 ).

**Figure 4 nutrients-14-01558-f004:**
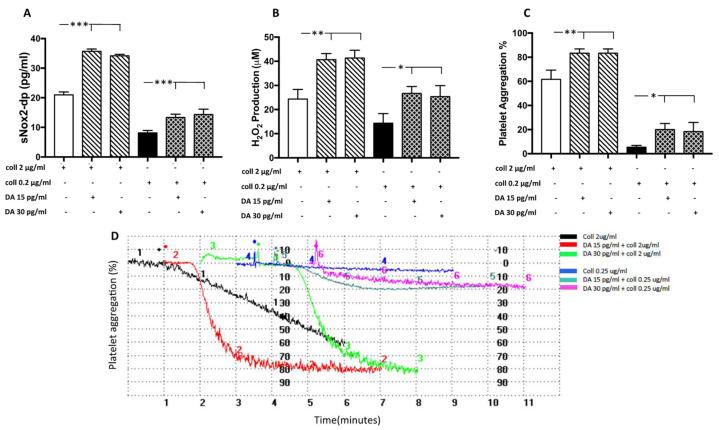
Dopamine-induced platelet aggregation and oxidative stress. sNOX2dp concentration (**A**), H_2_O_2_ levels (**B**) and the percentage of platelet aggregation (**C**) with relative representative tracing (**D**) in platelets stimulated with dopamine (DA, 15 and 30 pg/mL) in the presence of a sub-threshold (0.25µg/mL) or threshold (2 µg/mL) concentration of collagen (n = 3) (*** *p* < 0.001; ** *p* < 0.01, * *p* < 0.05).

**Figure 5 nutrients-14-01558-f005:**
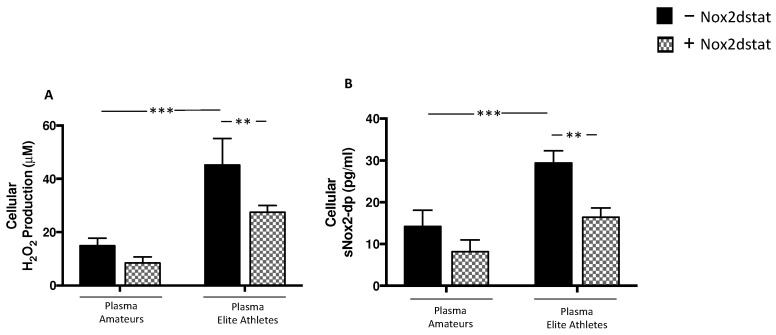
Oxidative stress in supernatants of HSMM-plasma co-cultures. H_2_O_2_ levels (**A**), and sNOX2dp concentration (**B**) were evaluated in supernatants of HSMM (Human Skeletal Muscle Myoblast) cell cultures treated with plasmas taken from amateurs (n = 3) and elite athletes (n = 3) in the presence or not of NOX2ds-tat. (n = 3) (Data are represented as mean ± standard deviation) (*** *p* < 0.001, ** *p* < 0.01).

**Figure 6 nutrients-14-01558-f006:**
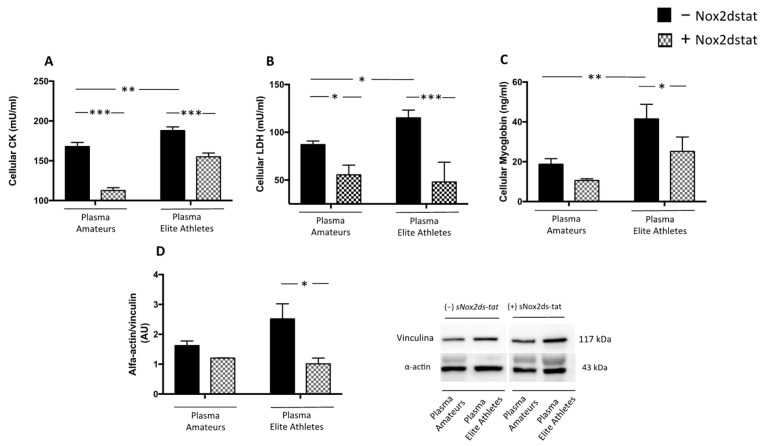
Muscle injury biomarkers in supernatants of HSMM-plasma co-cultures. Cellular CK (**A**), LDH (**B**) myoglobin (**C**) levels and α-actin expression (**D**) were evaluated in HSMM (Human Skeletal Muscle Myoblast) cell cultures treated with plasmas taken from amateurs (n = 3) and elite athletes (n = 3) in the presence or not of NOX2ds-tat. (n = 3) (Data are represented as mean ± standard deviation) (*** *p* < 0.001, ** *p* < 0.01, * *p* < 0.05).

**Figure 7 nutrients-14-01558-f007:**
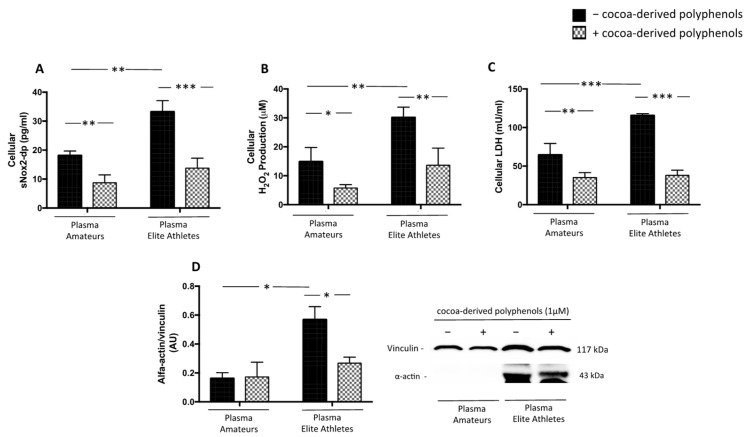
Oxidative stress and muscle injury biomarkers in supernatants of HSMM-plasma co-cultures. H_2_O_2_ levels (**A**), sNOX2dp concentration (**B**), LDH (**C**) and α-actin expression (**D**) were evaluated in supernatants of HSMM (Human Skeletal Muscle Myoblast) cell cultures treated with plasmas taken from amateurs (n = 3) and elite athletes (n = 3) in the presence or not of cocoa-derived polyphenols. (Data are represented as mean ± standard deviation) (*** *p* < 0.001, ** *p* < 0.01, * *p* < 0.05).

**Figure 8 nutrients-14-01558-f008:**
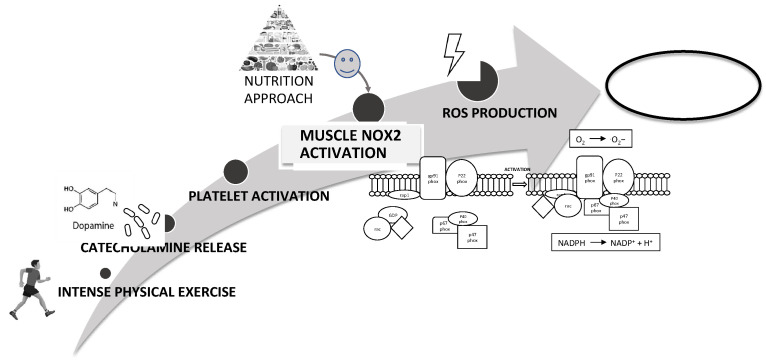
Schematic representation of intensive exercise-induced muscle injury via NOX2-mediated oxidative stress as a fundamental mechanism in the impairment of muscle cell integrity. A possible nutrition strategy using foods with antioxidant could reduce NOX2 activation and muscle injury improving athletic performance.

**Table 1 nutrients-14-01558-t001:** Baseline characteristics of Amateurs and elite football players.

	Amateurs Athletes (n = 23)	Elite Football Players (n = 23)	*p*
Age (years)	30.2 ± 4.7	30.1 ± 4.8	0.943
Gender (M/F)	23/0	23/0	-
WBC (×10^3^ μL)	7.2 ± 2.2	5.9 ± 1.4	0.02
PLT (×10^3^ μL)	215.3 ± 40.3	210.6 ± 49.5	0.725
RBC (×10^6^ μL)	5.1 ± 0.4	5.1 ± 0.39	0.99
Colesterol (mg/dL)	185.1 ± 30.8	172.3 ± 29.4	0.163
Glycaemia (mg/dL)	89.0 ± 28.8	83.5 ± 15.2	0.422
Height (cm)	179.8 ± 4.4	185.5 ± 5.6	<0.001
Weight (kg)	78.8 ± 6.8	81 ± 6.7	0.549
BMI	24.3 ± 1.9	24.3 ± 1.3	0.99
Heart rate at rest (bpm)	62.1 ± 10.4	56.3 ± 11.6	0.08
Systolic blood pressure (mmHg)	114.8 ± 11.2	111.7 ± 7.9	0.284
Diastolic blood pressure (mmHg)	74.1 ± 7.2	70.9 ± 6.8	0.128
Training per week (h)	5.1 ± 2.0	14.4 ± 1.1	<0.001
Sport practice (years)	12.7 ± 4.6	16 ± 1.2	0.002
Maximum workload (METs)	12.2 ± 1.8	15.4 ± 1.9	<0.001
Peak heart rate (bpm)	164.7 ± 6.9	169 ± 11.5	0.131
sCD40L (pg/mL)	220 [183–251]	294 [264–338]	<0.001
PDGF-bb (ng/mL)	4.9 [4.7–6.4]	8.1 [6.0–9.7]	<0.001
Dopamin (pg/mL)	8.1 [7.6–9.6]	24 [12–31]	<0.001
H_2_O_2_ (μM)	15 [10–19]	18 [13–32]	0.02
sNOX2dp (pg/mL)	1.5 [0.6–3.1]	8.6 [3.5–11]	<0.001
HBA (%)	44 ± 12	32 ± 6	<0.001
CK (mU/mL)	491 ± 62	669 ± 160	0.04
LDH (mU/mL)	109 [94–147]	152 [128–191]	<0.001
Myoglobin (ng/mL)	108 [96–111]	111 [98–161]	0.135

**Table 2 nutrients-14-01558-t002:** Correlation between biomarkers of muscle injury and oxidative stress and platelet activation.

	CK	
	Rs	*p* Value
sNOX2dp	0.112	*p* = 0.02
H_2_O_2_	0.016	*p* = 0.124
sCD40L	0.08	*p* < 0.05
PDGF-bb	0.268	*p* < 0.001
	**LDH**	
	**Rs**	** *p* ** **Value**
sNOX2dp	0.121	*p* = 0.017
	**Myoglobin**	
	**Rs**	** *p* ** **Value**
sCD40L	0.09	*p* < 0.05

**Table 3 nutrients-14-01558-t003:** Total Polyphenols Content in Dark Chocolate.

Compounds	Dark Chocolate
Total polyphenols, μg GAE/mL	799
Epicatechin, mg/g	0.65
Catechin, mg/g	0.26

## Data Availability

The data presented in this study are available on request from the corresponding author.

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
