# Peer review of "Platelet Activation Favours NOX2-Mediated Muscle Damage in Elite Athletes: The Role of Cocoa-Derived Polyphenols"

_nutrients, 2022, doi:10.3390/nu14081558_

Round 1
Reviewer 1 Report
The present study analyzed the differences of between elite athletes and amateur athletes, and the effect of cocoa-derived polyphenols on markers of oxidation. The manuscript is overall well written and presented. According to the authors, the purposes of the study were: 1) compare platelet activation, oxidative biomarkers and muscle damage induced by physical activity in amateurs and elite athletes; 2) exploit? if platelet activation and oxidative stress could lead to muscle damage; 3) verify if natural molecules with scavenging activity, i.e. polyphenols, could help in reducing muscle damage in vitro.
Please correct the verb in the second aim, it is likely to be “explore”
The most important comment is related to the cell culture section, which is missing some information. Please explain clearly the experimental design (controls, definition of time and dose for treatment), how polyphenols were added to the cell culture (vehicle) and how many biological and technical replicates of each experiment were conducted?
Please discuss the limitations of the study.
Correlations were not mentioned in the statistical analysis.
How the sample size was estimated?
Author Response
Reviewer 1
The present study analyzed the differences of between elite athletes and amateur athletes, and the effect of cocoa-derived polyphenols on markers of oxidation. The manuscript is overall well written and presented. According to the authors, the purposes of the study were: 1) compare platelet activation, oxidative biomarkers and muscle damage induced by physical activity in amateurs and elite athletes; 2) exploit? if platelet activation and oxidative stress could lead to muscle damage; 3) verify if natural molecules with scavenging activity, i.e. polyphenols, could help in reducing muscle damage in vitro.
Please correct the verb in the second aim, it is likely to be “explore”
Answer: Amended as suggested.
The most important comment is related to the cell culture section, which is missing some information. Please explain clearly the experimental design (controls, definition of time and dose for treatment), how polyphenols were added to the cell culture (vehicle) and how many biological and technical replicates of each experiment were conducted?
Answer: As suggested, we now added more information about the experimental design of the cell culture study (see page 6 lines 222-231).
Please discuss the limitations of the study.
Answer: As suggested, we now discuss the limitations of the study (see page 15 lines 501-514).
Correlations were not mentioned in the statistical analysis.
Answer: As suggested, we added correlation analysis in the Statistical Methods Section (see page 7 lines 271-281).
How the sample size was estimated?
Answer: As suggested, we added the sample size calculation in the Statistical Methods Section (see page 7 lines 285-289).
Reviewer 2 Report
Dear authors,
Please find attached my review of your article.

Author Response
Reviewer 2
First of all, I would like to inform you that your manuscript would definitely fit into Nutrients. However, important information is missing in your manuscript, so that the current version cannot be accepted. The most important aspects are explained below:
Abstract: Word limit of 200 was exceeded. Please shorten
Answer: Amended as suggested
Introduction:
The relationship between the first section and the overall research question is not clearly described. Furthermore, no recent previous research is described in detail. What evidence leads to the research question. Are there differences between amateur and elite athletes? The intent of the study is not clear.
Answer: Thank you to the referee. Now we better clarify this issue (see page 2 lines 58-69).
References for the statements are missing
in line: 66/67
in line: 78/79
in line: 81-83
in line: 87-90
in line: 92-95
Answer: We added new references as indicated (see references numbers: 8, 12, 13, 18, 19, 20, 22).
Materials and Methods
Study Population:
The most decisive factor is the description of the subject profile. The amateur athletes were described barely to insufficiently. What kind of sport do they do? General? Endurance or strength? Or rather sports, if so which ones. How many do which sports? What is the ratio of strength and endurance in the sports. Can I even compare them with football? Why only men and not women?
Answer: We are grateful to the reviewer for pointing out this important point. We modified the methods; accordingly, we already specified in the text that all amateur athletes practice mixed sports disciplines except football. In particular, 14 athletes played tennis/padel tennis, 4 athletes played volleyball, 4 athletes played basketball and 1 practiced surfing. We decided to include only mixed sports, which fall in the same category as football, to avoid more confounding factors. Based on the Dal Monte Lubich sports classification, which refers to the energy characteristics of the sports, football is a sport discipline characterized by an alternated aerobic - anaerobic impact. The sports practiced by the amateur athletes fall in the same category. Similarly, in the cardiovascular classification of the Olympic sport disciplines, football and the other disciplines falls in the category of the "mixed sports" which demonstrate a balanced cardiovascular remodelling, according to the relative isometric and isotonic components of exercise and resulting cardiovascular adaptation. Most mixed sports practiced by the amateur athletes, as tennis and padel tennis are characterized by a high-intensity intermittent activity, which combines high-frequency (0.7–1.5 per second (s)) and low-intensity actions during rallies that are of a moderate duration (9–15 s), interspersed by 1020 s of rest in between, leading to longer breaks of 90s.
Only male athletes were enrolled in this study because platelet activation and oxidative stress are influenced by sexual hormonal status, as demonstrated by a recent publication from our group (Raparelli V, J Endocrinol Invest. doi: 10.1007/s40618-022-01771-0.) so, we decided a priori to include only male sex to avoid confounding factors related to the hormonal cycle. We added this last issue in limitation of the study (see page 3 lines 113-115; lines 131-140; page 15, lines 505-510, and Table 1).
Elementary information is also missing in the description of the footballers' training hours. When were the blood samples taken. In the preparation phase? during the season. directly after exercise? Which players? Regular players? Substitute players etc. (no regular playing times).
Answer: We thank the reviewer for pointing out these observations. We added the description of the habitual training hours, specifically during the training and competitive seasons, elite athletes were engaged in a 120-minute training (including a 15-minute warm-up, 30- minute technical-tactical skills, 30-minute aerobic training reaching 75% of the maximal heart rate, 30-minute strength training, and 15-minute cooldown) 6 times per week and a 90-minute match per week. The elite athletes were all primary division (Italian ‘‘Serie A’’ team) members of the first-league A.S. Roma Football team, they trained at least 15h/week, >6.0 METs, and had at least 8 years of competitive experience. In addition, we added a new paragraph dedicated to blood sampling and preparation. Blood samples were collected prior to the beginning of the training season, at the same time of the pre-participation screening (see page 3 lines 113-116 and lines 118-124; page 5 lines 157-172).
Football training can also have different focal points, such as conditional or tactical components. What was the ratio of the different sport-specific training components during the intervention period?
Answer: As the intervention period was immediately prior to the beginning of the training and competitive season (at the time of the pre-participation screening), the athletes did not specifically trained with training programs, therefore we cannot answer to this question as this is not pertinent to the timing of the blood samples. The training session was not standardized for the elite nor for the amateur athletes, as the elite athletes were enrolled immediately prior to the beginning of the training and competitive season, after 45 days off-season holidays in which they didn’t carry out specific activities such as soccer but only recreational sports fitness. We added this issue in the limitation of the study (see page 15 lines 510-511).
In addition, information on the last sport unit of all participants before the blood sample is missing.
Answer: Thank you for this insight, we specified in the added paragraph dedicated to blood sampling and preparation that "All elite and amateur athletes abstained from training for 24h before blood sampling, as requested by our study protocol." (see page 5 lines 157-172).
This information is essential to compare the blood values with each other.
2.1.3 Evaluation of oxidative stress
why only H2O2? There are several other oxygen radicals? Is it not possible that other oxygen radicals (for example superoxide or peroxide radicals) are produced primarily?
I request clarification
Answer: We agree with the referee that other oxygen radicals, and in particular superoxide (O2- ), are produced primarily by NOX2. However, we evaluated H2O2 as, among all the reactive oxygen species, is a longer-lasting signaling molecule, whose passive diffusion across the lipid bilayer of biological membranes can occur (Di Marzo N et al. Cells. 2018;7(10):156). Therefore, H2O2 has a lifetime and typical concentration higher than the superoxide anion. Finally, H2O2 possesses the ability to greatly contribute to oxidative stress by amplifying NOX2 activation (El Jamali A et al. Free Radic Biol Med. 2010;48:798–810).
2.3 Statistical Methods
Statistical analysis is not sufficiently described. Was there no check for normal distribution? Individual biomarkers such as CK show a high variability, so that their statistical evaluation is carried out with the logarithmised values. Based on the results of the normal distribution, the individual parameters are either calculated with a parametric or with the non-parametric tests.
Answer: Thank you to the referee. We added a more detailed description of the statistical analysis performed. In particular, we performed non-parametric tests (Mann Whitney U test) for the variables that did not have a normal distribution (calculated through the Shapiro-Wilk Test; furthermore, we added the effect size and the sample size calculation (see page 7 lines 271-277).
What effects could be observed? Effect size calculation?
Answer: In this final version of the manuscript, we added the effect size calculation for s-NOX2-dp (see the paragraph “Effect size and sample size determination”) (see page 7 lines 283-284).
Results:
3.1 in vivo study
What is the time interval between the last training session and the blood sample for elite and amateur athletes? Was the training session standardised for the amateur athletes? Here again it is crucial what type of athlete has been recruited.
In line 235: intense exercise? What is the time interval between the last training session and the blood sample for elite and amateur athletes? Was the training session standardised for the amateur athletes? Here again it is crucial what type of athlete has been recruited.
Answer: Thank you for this insight, we specified in the added paragraph dedicated to blood sampling and preparation that "All elite and amateur athletes abstained from training for 24h before blood sampling, as requested by our study protocol." The training session was not standardized for the elite nor for the amateur athletes, as the elite athletes were enrolled immediately prior to the beginning of the training and competitive season, after 45 days off-season holidays in which they didn’t carry out specific activities such as soccer but only recreational sports fitness (see page 5 lines 157-172).
3.1.3 Intensive exercise induces elevation of specific muscle enzymes as creatine kinase is a parameter that can be influenced very much. Therefore for all parameters the individual values should also be shown in the graphs.
Answer: As suggested, we now showed individual values in graphs (see Figures 1-3).
4 Discussion:
In Line 343: The statement cannot be made with the data on amateur athletes described above. Decisive aspects of the requirement profile of amateur athletes were not explained. A decisive aspect is which sports are practised on average 5h a week. Based on the lack of information in the methods, most of the statements in the discussion cannot currently be left as they are. Furthermore, aspects of the different effects of different sports are missing.
Answer: As suggested, we added this information in the text (see page 3 lines 131-140 and Table 1).
In Line 427: No limitations of the work were mentioned. The most important limiting factor is to which sport the elite footballers are compared. In order to make a comparison, it primarily makes sense to compare similar sports such as football with the elite athletes. Then the different effects can be attributed to the training frequency and duration. Another aspect that was not addressed is whether or not similar effects can be expected for women. Do elite female footballers also have similar levels of oxidative stress?
Answer: As suggested we added a paragraph about the limitation of the study (see page 15 lines 501-514).

Round 2
Reviewer 1 Report
The authors addressed all the comments correctly.
Reviewer 2 Report
Dear authors,
I like the revised version of your manuscript. You have addressed all my questions and comments in detail. Therefore, I have no further comments and your manuscript can be published in my opinion.
This manuscript is a resubmission of an earlier submission. The following is a list of the peer review reports and author responses from that submission.